# Factors Affecting Nurses’ Health Promotion Behavior during the COVID-19 Pandemic Based on the Information–Motivation–Behavioral Skills Model

**DOI:** 10.3390/medicina58060720

**Published:** 2022-05-27

**Authors:** Sun-gyung Lee, Boyoung Kim

**Affiliations:** 1Gyeongsang National University Hospital, Jinju 52727, Korea; leipe@naver.com; 2Department of Nursing, College of Nursing, Chonnam National University, Gwangju 61469, Korea

**Keywords:** COVID-19, health promotion, knowledge, attitude, infection, social support, self-efficacy, secondary trauma

## Abstract

*Background and Objectives:* The COVID-19 pandemic has emerged as a major threat to nurses’ health. This study aimed to investigate the factors affecting nurses’ health promotion behaviors during the coronavirus disease 2019 (COVID-19) pandemic. *Material and Methods:* The participants were clinical nurses who had direct contact with patients at a university hospital in G province, Korea. Data were collected from March 16 to April 16, 2021, and the final analysis included data from 162 nurses. The general and lifestyle characteristics of the participants were analyzed using descriptive statistics, and the effect on health promotion behavior was analyzed using multiple regression with SPSS/WIN 21.0. *Results:* The results showed that the factors influencing nurses’ health promotion behavior during the COVID-19 pandemic were social support (β = 0.40, *p* < 0.001), self-efficacy (β = 0.27, *p* = 0.014), being married (β = 0.18, *p* = 0.018), having good health (β = 0.31, *p* < 0.001), and not skipping meals (β = 0.20, *p* = 0.001). The explanatory power of the variables was 51.4%. Therefore, health promotion programs to promote social support and self-efficacy are needed to improve nurses’ health promotion behaviors during the COVID-19 pandemic. *Conclusions:* These results indicate that the development of additional management strategies for health promotion among nurses during the COVID-19 pandemic is necessary. It is necessary to prepare organizational policies and manage self-care to improve nurses’ irregular eating habits during the ongoing pandemic.

## 1. Introduction

The coronavirus disease 2019 (COVID-19) is a new infectious disease that was first reported in Hubei Province, China, in December 2019 [1]. Global confirmed cases have surpassed 3.4 billion and new variants of the disease continue to emerge [2]. As the resurgence of COVID-19 accelerates worldwide, countries are facing multiple challenges in responding to the pandemic due to difficulties in core quarantine requirements, such as fast diagnosis, patient isolation, and contact management [3]. Thus, the spread of local outbreaks and the increase in deaths are expected to continue for a while [1,2,3].

Health care workers are responsible for attending to suspected and confirmed cases of COVID-19 [4]. Nurses, in particular, are highly vulnerable to infection because they have the most direct contact with patients and caregivers. Furthermore, the associated psychological difficulties can affect their health promotion behavior [5]. In particular, frontline nurses who care for and stay in close contact with suspected and confirmed COVID-19 cases experience physical and psychological distress due to sleep disorders, anxiety, depression, and post-traumatic stress [6]. Nurses are role models for patients; hence, their health promotion is significant as it affects both their own and their patients’ lifestyles [7].

The information–motivation–behavioral skills (IMB) model is widely used to understand and promote health-related behavior [8]. The IMB model includes three primary constructs that influence behavioral changes: information and knowledge about the behavior, an individual’s motivation to perform the behavior, and the behavioral skills necessary to perform the behavior [9]. Being informed is directly related to and is a significant determinant of health-related behaviors [8,9]; thus, it is crucial in health promotion behaviors, such as disease prevention [10]. A study on the factors affecting the health promotion behavior of university students reported that knowledge had a strong and positive correlation with health promotion behavior and that providing knowledge was essential for changing health perception and behavior [11]. Therefore, understanding and using correct information on health practices and infection prevention are necessary to promote health-related behaviors during the COVID-19 pandemic.

Motivation is another determinant of health promotion behavior and can be divided into personal and social motivation [9,12]. Personal motivation refers to individual attitudes and beliefs, and social motivation is the perception of and social support for subjective norms [13]. A previous study on nurses reported that attitudes affect preventive health promotion behaviors [14]. In particular, the attitude of frontline nurses caring for patients with COVID-19 may influence their health promotion behavior. Among social motivations, social support is a positive resource that an individual can obtain from their relationships [15], with many health care workers reporting that they could endure and overcome difficulties in fighting COVID-19 through the support of family members and colleagues [16]. Numerous studies have also concluded that social support is a major variable influencing health promotion behavior.

In the IMB model, behavioral skills refer to the confidence that an individual can successfully perform the behavior required to produce a specific outcome with self-efficacy [17]. Even information-savvy and highly motivated individuals need skills and the confidence to practice them on their own to effectively perform preventive and management behaviors in daily life [18]. Therefore, self-efficacy is closely related to health promotion behavior, and the higher the self-efficacy, the better the health promotion behavior [19]. A study on shift workers reported that self-efficacy directly motivates health promotion behavior and affects the continuation of the behavior [20]. Therefore, self-efficacy is a significant factor for nurses to engage in health promotion behaviors successfully during the COVID-19 pandemic.

Meanwhile, the long-term impact of COVID-19 can lead to chronic stress, and nurses caring for COVID-19 patients may experience traumatic events [21]. The frequency and intensity of exposure to such events can have a significant impact on post-traumatic stress [22]. This secondary traumatic stress can affect nurses’ health promotion behavior and result in adverse effects on patients, such as neglecting or avoiding patients’ needs [23]. However, there is a lack of research on the impact of these psychological factors on nurses’ health promotion behaviors as the COVID-19 pandemic continues. Therefore, this study model sets the COVID-19 pandemic, a specific environmental and personal characteristic factor, and secondary traumatic stress as independent variables that can affect health promotion behavior. Each factor can have a direct or indirect effect, but this study assumes that they directly affect health promotion behaviors.

Based on these assumptions, this study aimed to provide basic data to improve nurses’ health promotion behavior by applying the IMB model to identify factors affecting their health promotion behavior during the COVID-19 pandemic. The detailed purposes of this study are as follows:(1)Identify knowledge of COVID-19, attitude toward COVID-19 infection, social support, self-efficacy, secondary traumatic stress, and health promotion behavior of nurses.(2)Identify the differences in health promotion behaviors according to the general and lifestyle characteristics of nurses.(3)Identify the correlation between nurses’ knowledge of COVID-19, attitude toward COVID-19 infection, social support, self-efficacy, secondary traumatic stress, and health promotion behavior.(4)Identify the factors affecting health promotion progress among nurses.

## 2. Method

### 2.1. Research Design

This descriptive correlational study investigated the factors affecting nurses’ health promotion behaviors.

### 2.2. Participants and Data Collection

The participants were clinical nurses working at a tertiary university hospital in G City, Korea. The hospital serves as a medical center for the entire western Gyeongsangnam-do region of Korea, with numerous general hospitals clustered around it. The hospital has approximately 900 beds, and the staff comprised 313 doctors and 954 nurses at the time of this study. Data were collected from March 16 to April 16, 2021, in compliance with the social distancing measures and quarantine guidelines of Korea. The criteria for the selection of participants for this study were as follows: (1) nurses working at a tertiary general hospital, (2) nurses ranked below the head nurse, and (3) nurses who had been working in the hospital for 12 months. Only those who read the guidelines and agreed to participate in this study took part. The minimum sample size, including six predictors, was 146 nurses, based on a significance level (α) of 0.05, a power (1-β) of 0.95, and an effect size of 0.15 for a two-sided multiple linear regression analysis using G*power 3.1.7 (Heinrich Heine University, Düsseldorf, Germany). Considering a 20% dropout rate, 175 copies were distributed and 164 copies were collected. The final analysis used data from 162 copies, excluding two with incomplete answers.

### 2.3. Research Tools

The measurement tools were self-report questionnaires on general characteristics, knowledge of COVID-19, attitude toward COVID-19 infection, social support, self-efficacy, secondary traumatic stress, and health promotion behavior.

#### 2.3.1. Sociodemographic and Lifestyle Characteristics

Sociodemographic characteristics (i.e., gender, age, marital status, family to take care of, educational level, disabilities or health problems, and perceived health status) and lifestyle characteristics (i.e., walking, sleep disruptions, number of times meals are skipped, stress management) were collected using a structured questionnaire.

#### 2.3.2. Knowledge about COVID-19

Knowledge about COVID-19 was measured by modifying and supplementing the prevention guidelines of the tool developed by Yoon [24]. Example items included the following: “COVID-19 vaccines have not been developed yet,” “For upper respiratory tract specimens, either an oropharyngeal swab or a nasopharyngeal swab is collected and placed in a single virus receiving medium,” and “Patients are released from quarantine when 48 h have elapsed after all their symptoms have disappeared and when all PCR test results are negative (conducted twice, 24 h apart).” The final tool consisted of 21 questions. Scores ranged from 0 to 21 (an incorrect answer received 0 and a correct answer received 1), with a higher score indicating a higher knowledge level. The reliability of this study was KR = 0.71.

#### 2.3.3. Attitude toward COVID-19 Infection

Attitude toward COVID-19 infection refers to the feeling or perception with regard to COVID-19. This was measured using a tool developed by Park [25], which was modified and supplemented by Choi and Lee [26]. It consists of 12 items scored on a five-point Likert scale (1 = strongly disagree, 5 = strongly agree). The higher the score, the more positive the attitude toward COVID-19 infection, which meant greater compliance with COVID-19 response guidelines. In this study, Cronbach’s α for this tool was 0.80.

#### 2.3.4. Social Support

Social support refers to positive resources that an individual can derive from interpersonal relationships. Social support was measured using a tool adapted and modified by Yang [27] based on the Social Provisions Scale developed by Cutrona and Russell [28]. It consists of 20 items scored on a five-point Likert scale (1 = strongly disagree, 5 = strongly agree). The higher the score, the higher the social support from colleagues. In this study, Cronbach’s α for this tool was 0.94.

#### 2.3.5. Self-Efficacy

Self-efficacy refers to the belief that an individual can successfully perform the behavior required to produce a certain outcome. Self-efficacy was measured using a tool modified and supplemented by Jung [29], based on the self-efficacy scale developed by Sherer and Maddux [30]. It consists of 17 items scored on a five-point Likert scale (1 = strongly disagree, 5 = strongly agree). In this study, Cronbach’s α for this tool was 0.95.

#### 2.3.6. Secondary Traumatic Stress

Secondary traumatic stress is a measure of the impact on an individual who has indirectly experienced trauma. Secondary traumatic stress was measured using the Korean version of the secondary traumatic stress scale adapted and modified by Park [31], based on the original version developed by Bride et al. [32]. It consists of 13 items scored on a five-point Likert scale (1 = strongly disagree to 5 = strongly agree). The higher the score, the higher the secondary traumatic stress. In this study, Cronbach’s α for this tool was 0.88.

#### 2.3.7. Health Promotion Behavior

Health promotion behavior refers to a positive approach toward health for maintaining or enhancing the optimal well-being of an individual or group, self-actualization, and individual self-esteem. Health promotion behavior was measured using the tool developed by Kim et al. [33]. It consists of 15 items scored on a five-point Likert scale (1 = strongly disagree, 5 = strongly agree). The higher the score, the higher the level of health promotion behavior. In this study, Cronbach’s α for this tool was 0.82.

### 2.4. Data Analysis Method

The general and lifestyle characteristics of the participants were analyzed using descriptive statistics with SPSS/WIN 21.0 (IBM Corp, Armonk, NY, USA). The differences in health promotion behavior according to general and lifestyle characteristics were confirmed by an independent *t*-test and one-way analysis of variance. Post hoc testing was performed using Scheffé’s test. Variables affecting health promotion behavior were examined using Pearson’s correlation coefficient and multiple regression analysis.

### 2.5. Ethical Considerations

This study was conducted after obtaining approval from the Institutional Review Board of G-University Hospital in G-Province (approval number: GNUH 2021-01-023-002). Informed consent was obtained from the head of the participating hospital after explaining the purpose and procedures of this study. The participants who completed the questionnaires received small gifts.

## 3. Results

### 3.1. Participant Characteristics

#### 3.1.1. General Characteristics

A total of 162 participated in this study. The mean and standard deviation of health promotion behavior scores are shown in Table 1. Of these, 155 (3.41 ± 1.11) were women and 7 (3.30 ± 0.53) were men. In terms of education, 147 (3.31 ± 0.57) had a college degree or lower, and 15 (3.29 ± 0.45) had a master’s degree or higher. In terms of health problems, 118 (3.26 ± 0.53) participants answered that they had health problems, whereas 44 (3.45 ± 0.62) said that they were healthy. We further inquired about the specific disability or health problem that these 118 participants had and found the following: neck/back pain (77), sleep disorders (40), emotional problems (15), eye/vision problems (14), and other (13) (see further disabilities or health problems in Appendix A). In terms of the number of hospital visits per month, 58 participants answered that they visited the hospital at least once per month. An independent-samples *t*-test was performed to verify the differences in health promotion behaviors according to participant characteristics. The general characteristics that showed significant differences in health promotion behavior were marital status (t = −2.04, *p* = 0.043) and health status (t = 3.89, *p* < 0.001).

#### 3.1.2. Lifestyle Characteristics of the Participants

Table 2 shows the lifestyle characteristics of the participants. In terms of physical activity (walking for more than 20 min), 40 (3.48 ± 0.58) participants reported that they walked more than thrice a week, 70 (3.34 ± 0.51) walked less than twice a week, and 52 (3.13 ± 0.57) did not walk for the given duration at all. A total of 39 participants (3.41 ± 0.60) reported no sleep disruptions, 70 (3.32 ± 0.54) experienced sleep disruptions less than twice a week, and 53 (3.22 ± 0.56) reported sleep disruptions more than three times a week. With regard to the frequency of skipping meals, 17 (3.60 ± 0.66) answered never, 58 (3.39 ± 0.55) answered less than twice a week, and 87 (3.19 ± 0.52) chose more than three times a week. Regarding lack of rest, 15 participants (3.57 ± 0.70) answered that they had no lack of rest, 46 (3.37 ± 0.50) answered less than twice a week, and 101 (3.24 ± 0.55) answered more than three times a week. In terms of stress management, 98 (3.23 ± 0.57) were unable to manage stress at all, 47 (3.35 ± 0.49) answered less than twice a week, and 17 (3.64 ± 0.58) chose more than three times a week. The lifestyle characteristics that showed differences in health promotion behavior were walking (F = 4.91, *p* = 0.043), number of meals skipped (F = 5.06, *p* = 0.007), and stress management (F = 4.18, *p* = 0.017).

### 3.2. The Correlations between Knowledge about COVID-19, Attitude toward COVID-19 Infection, Social Support, Self-Efficacy, Secondary Traumatic Stress, and Health Promotion Behavior

Table 3 shows the correlations between knowledge about COVID-19, attitude toward COVID-19, social support, self-efficacy, secondary traumatic stress, and health promotion behavior. Attitudes toward COVID-19 infection (r = 0.38, *p* < 0.001), self-efficacy (r = 0.54, *p* < 0.001), and secondary traumatic stress (r = 0.53, *p* < 0.001) were significantly correlated with health promotion behaviors.

### 3.3. Factors Affecting the Participants’ Health Promotion Behavior

A multiple regression analysis was conducted to examine the factors affecting nurses’ health promotion behavior during the COVID-19 pandemic. Table 4 shows the results. The Durbin–Watson statistic was used to check for autocorrelation between error terms before the analysis; the results showed that there was no autocorrelation. A normal distribution was assumed for the error terms. On examining whether there is multicollinearity between the independent variables through the tolerance limit and the variance inflation factor, the tolerance limit was 0.66–0.90 (higher than 0.10), and the variance inflation factor was 1.09–1.57 (less than 10). Thus, there was no multicollinearity between the independent variables.

According to the analysis, the variables affecting health promotion behavior were social support (β = 0.40, *p* < 0.001) and self-efficacy (β = 0.27, *p* = 0.014). In other words, higher social support and self-efficacy were correlated with better health promotion behavior.

The adjusted R2 was 0.514, meaning that the measured variables explained 51.4% of the variance in health promotion behavior.

## 4. Discussion

This study aimed to understand the effects of knowledge about COVID-19, attitudes toward COVID-19 infection, social support, secondary traumatic stress, and self-efficacy on the health promotion behavior of nurses during the COVID-19 pandemic by applying the IMB model. After analyzing the factors influencing health promotion behavior, it was found that social support, self-efficacy, being married, good health, and not skipping meals had positive effects on health promotion behavior; the model’s explanatory power was 51.4%. Based on these results, the main findings are summarized as follows:

Among the factors influencing health promotion behavior, social support had the greatest influence on health promotion behavior; the higher the social support, the better the health promotion behavior. These results are consistent with those of other studies reporting that social support affects health promotion behavior among various groups, including correctional officers [34] and middle-aged adults [35]. A study on the relationship between social support and health promotion behavior among hospital nurses [36] also reported that social support had a positive effect on health promotion behavior, which supports the results of this study.

A survey on health care workers (COVID-19 field response teams) also found that the social support of colleagues was the driving force that allowed these workers to continue their work [34]. Frontline nurses caring for patients during the COVID-19 pandemic may experience social isolation, face tremendous responsibility, and suffer from psychological atrophy [5]. These psychological difficulties may lead to neglect of health promotion behavior. Therefore, the circumstances created by the COVID-19 pandemic should be considered to promote the health promotion behavior of nurses. In addition, as the social support of frontline nurses can affect patients and promote the health of individual nurses, the use of social support resources should also be considered.

After social support, self-efficacy had the greatest influence on health promotion behavior, and the higher the self-efficacy, the better the health promotion behavior. Studies on nurses [37] and nursing students [38] have also reported similar results, confirming that self-efficacy is a major factor that promotes health promotion behavior. A study on the factors affecting health promotion behavior among shift workers [20] reported that those who received health-related education exhibited high self-efficacy. In addition, the risk of receiving inaccurate information about COVID-19 lowered nurses’ self-efficacy [39]. Therefore, to increase nurses’ self-efficacy, sharing accurate knowledge and information about novel infectious diseases such as COVID-19 is crucial to prevent infections in hospitals and promote nurses’ health promotion behavior. Self-efficacy is also essential for self-care; therefore, it should be improved through health education.

Attitude toward COVID-19 infection was significantly correlated with health promotion behavior but was not an explanatory factor. A study of college students [40] reported that health beliefs about emerging infectious diseases had a significant influence on hygiene behavior, which was different from the results of this study. The results may not be consistent because this study had different participants, used other tools to measure health promotion behavior, and focused on the COVID-19 pandemic. Hence, these factors should be considered when interpreting the results. It is also necessary to test for mediation effects because other studies have reported that personal motivation mediates self-efficacy [40,41].

This study expanded the IMB model’s conceptual framework to set the COVID-19 pandemic as a specific scenario and added secondary traumatic stress, a psychological factor, among the personal characteristic factors. However, although this study expected high secondary traumatic stress and a negative impact on health promotion behavior among nurses due to the risks of isolation and direct infections due to COVID-19, secondary traumatic stress was not a statistically significant factor. Although secondary traumatic stress was not statistically significant, the COVID-19 pandemic is still ongoing. Nurses may have an increased chance of developing secondary traumatic stress because they experience burnout due to excessive workload and perceive high stress due to direct contact with COVID-19 patients [40]. These factors can adversely affect their quality of life and health [42]. Therefore, in-depth studies should be conducted on the psychological factors that affect nurses’ health promotion behavior.

After investigating the general characteristics influencing the health promotion behavior of participants, married people showed better health promotion behavior than those who were single. This finding is consistent with that of a study on the health promotion behavior of general hospital nurses [43]. A study on shift-work nurses also reported similar results [44]. In the case of married people, domestic stability has a positive effect on their work [45]. They also have more people to provide social support and are more responsible for their health due to factors such as pregnancy, childbirth, and childcare, resulting in them expending more effort into health promotion behavior. However, other studies on the psychological impact of the COVID-19 outbreak on nurses have reported a high level of anxiety about the risk of infection due to concerns for family members [7,46]. These psychological difficulties of married people may promote infection prevention behavior and affect the performance of health-related behaviors during the outbreak of infectious diseases.

In addition, healthier nurses showed better health promotion behaviors than those who were unhealthy. A study on nurses working in shifts [44] also reported that better health promotion behavior resulted in better health conditions, indicating that health status is a factor influencing health promotion behavior. Therefore, the healthier one is, the better one can perform activities to maintain and promote health [9].

Finally, nurses who did not skip meals showed better health promotion behaviors. A study on college students [47] reported that those who did not skip breakfast showed health promotion behavior, similar to what was found in this study. Research has also shown that nurses lead irregular lives due to the nature of working three shifts and often skip meals due to excessive work [48]. Furthermore, during the COVID-19 pandemic, nurses have been overwhelmed by heavy workload and have developed irregular eating habits due to fatigue and stress [49]. Thus, mealtimes and diets provided by hospitals should be improved by considering their shift patterns and high-intensity infection control measures [50].

In summary, the most influential factors affecting nurses’ health promotion behaviors during the COVID-19 pandemic were social support, self-efficacy, and eating habits. Therefore, programs to promote social support and self-efficacy and measures to compensate for irregular eating habits are necessary to improve nurses’ health promotion behavior.

Despite its various strengths, this study has several limitations. The participants in this study were limited to nurses from one university hospital; hence, the results may not be applicable to all nurses. Follow-up studies should include more nurses working at other hospitals. Second, further research should validate the effects of developing and implementing health promotion programs, including the improvement of self-efficacy and support from colleagues, which were the factors influencing health promotion behavior in this study. Third, the explanatory power between secondary traumatic stress and health promotion behaviors was not significant in this study. In-depth research should be conducted on the influence of individual characteristics or psychological factors on health promotion behavior in the future. Fourth, this study applied the COVID-19 pandemic as a specific situation in the IMB model, but it failed to verify whether knowledge about COVID-19 and attitude toward COVID-19 influenced health promotion behavior through self-efficacy as a medium. Therefore, further research is needed.

## 5. Conclusions

This study investigated the factors affecting the health promotion behavior of nurses who have influenced patients during the COVID-19 pandemic through their roles as educators and caregivers. The results showed that social support, self-efficacy, marital status, perceived good health, and regular or correct eating patterns were the influential variables. In addition to caring for patients, frontline nurses attending to COVID-19 patients should maintain self-care. Therefore, hospitals should develop health promotion programs to increase the social support and self-efficacy of nurses based on the results of this study. It is also necessary to motivate nurses to develop a healthy lifestyle during the ongoing pandemic.

## Figures and Tables

**Table 1 medicina-58-00720-t001:** Participant Characteristics and Their Relationship to Health Promotion Behavior (N = 162).

Variables	Categories	*n*	Health Promotion Behavior Score (M ± SD)	t	*p*
Gender	Male	7	3.41 ± 1.11	0.26	0.801
	Female	155	3.30 ± 0.53		
Age (Yrs)	≤29	107	3.30 ± 0.54	−0.13	0.900
	≥30	55	3.32 ± 0.59		
Marital status	Single	126	3.26 ± 0.55	−2.04	0.043
	Married	36	3.48 ± 0.57		
Number of people living together	≤2	90	3.29 ± 0.58	−0.56	0.578
	≥3	72	3.34 ± 0.53		
Family to take care of	Yes	11	3.30 ± 0.45	−0.05	0.958
	No	151	3.31 ± 0.57		
Educational level	College	147	3.31 ± 0.57	0.16	0.874
	Master’s or above	15	3.29 ± 0.45		
Health problems	Yes	118	3.26 ± 0.53	−1.80	0.076
	No	44	3.45 ± 0.62		
Hospital use	No	103	3.37 ± 0.53	1.86	0.065
	≥1	58	3.20 ± 0.59		
Perceived health status	Healthy	95	3.45 ± 0.56	3.89	<0.001
	Less healthy than average	67	3.11 ± 0.50		

M = mean; SD = standard deviation.

**Table 2 medicina-58-00720-t002:** Lifestyle Characteristics of the Participants (N = 162).

Variables	Categories	*n*	Health Promotion Behavior Score (M ± SD)	t	*p*	Scheffé
Walk for more than 20 min	Not at all ^a^	52	3.13 ± 0.57	4.91	0.009	c > a
	Less than twice a week ^b^	70	3.34 ± 0.51			
	More than three times a week ^c^	40	3.48 ± 0.58			
Sleep disruptions	Not at all ^a^	39	3.41 ± 0.60	1.22	0.297	
	Less than twice a week ^b^	70	3.32 ± 0.54			
	More than three times a week ^c^	53	3.22 ± 0.56			
Number of times meals are skipped	Not at all ^a^	17	3.60 ± 0.66	5.06	0.007	a > c
	Less than twice a week ^b^	58	3.39 ± 0.55			
	More than three times a week ^c^	87	3.19 ± 0.52			
Lack of rest	Not at all ^a^	15	3.57 ± 0.70	2.54	0.082	
	Less than twice a week ^b^	46	3.37 ± 0.50			
	More than three times a week ^c^	101	3.24 ± 0.55			
Stress management	Not at all ^a^	98	3.23 ± 0.57	4.18	0.017	c > a
	Less than twice a week ^b^	47	3.35 ± 0.49			
	More than three times a week ^c^	17	3.64 ± 0.58			

M = mean; SD = standard deviation; ^a,^
^b,^
^c^ = Scheffe’s post hoc test.

**Table 3 medicina-58-00720-t003:** Correlations Between Health Promotion Score and each of the Study Variables.

Variables	KC	ACI	SS	SE	ST
	*r (p)*				
Health promotion behavior	−0.12 (0.127)	0.38 (<0.001)	−0.13 (0.135)	0.54 (<0.001)	0.53 (<0.001)

KC = knowledge about COVID-19; ACI = attitude toward COVID-19 infection; SS = social support; SE = self-efficacy; ST = secondary traumatic stress.

**Table 4 medicina-58-00720-t004:** Factors Affecting the Participants’ Health Promotion Behavior (N = 162).

	B	SE	β	t	*p*
(constant)	0.63	0.45		1.41	0.160
Marital status (1 = married, 0 = single)	0.16	0.07	0.13	2.13	0.035
Perceived health status (1 = healthy, 0 = less healthy than average)	0.13	0.06	0.13	2.27	0.024
Walk for more than 20 min (ref = not at all)					
Less than twice a week	0.06	0.07	0.06	0.81	0.420
More than three times a week	0.04	0.08	0.03	0.46	0.649
Number of times meals are skipped (ref = more than three times a week)					
Not at all	0.21	0.10	0.13	2.20	0.029
Less than twice a week	0.06	0.06	0.06	0.91	0.364
Stress (ref = not at all)					
Less than twice a week	0.08	0.07	0.07	1.15	0.252
More than three times a week	0.29	0.10	0.18	2.95	0.004
Knowledge about COVID-19	−0.70	0.36	−0.11	−1.91	0.058
Attitude toward COVID-19 infection	0.15	0.08	0.12	1.91	0.059
Secondary traumatic stress	−0.01	0.04	−0.01	−0.20	0.840
Social support	0.37	0.06	0.40	5.85	<0.001
Self-efficacy	0.23	0.05	0.27	4.31	<0.001
Adj-R^2^	0.514				
R^2^	0.553				
F(p)	14.09 (<0.001)				

## Data Availability

The data presented in this study are available from the corresponding author upon request due to privacy restrictions.

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
