# Peer review of "Factors Affecting Nurses’ Health Promotion Behavior during the COVID-19 Pandemic Based on the Information–Motivation–Behavioral Skills Model"

_medicina, 2022, doi:10.3390/medicina58060720_

Round 1
Reviewer 1 Report
Dear Authors,
I would like to congratulate you on your work.
I have just minor observations:
- Verify that your Keywords are according to the MeSH system
- Lines 93, 94: The aims of the study. For the aims, you should conceive the aims as a list (now you have paragraphs), either with "-" or maybe numbers for each aim, such as (1), (2), etc.
- In the Discussion, it is not necessary to start the paragraphs with "first..., second..., third...". It will be a more pleasant read
Author Response
Thank you for the opportunity to revise our manuscript. We appreciate this careful review and your constructive suggestions. The manuscript has improved substantially after making the suggested edits.
Dear Editors:
Thank you for the opportunity to revise our manuscript. We appreciate this careful review and your
constructive suggestions. The manuscript has improved substantially after making the suggested edits.
The detailed responses to each comment follow below.
Response to Reviewer 1 Comments
1. Verify that your Keywords are according to the MeSH system
Reply: Thank you for your comment. We have revised the keywords (shown in red font) based on your
suggestion as follows:
Keywords: COVID-19; health promotion; knowledge; attitude; infection; social support; self-
efficacy; secondary trauma
2. Lines 93, 94: The aims of the study. For the aims, you should conceive the aims as a list (now
you have paragraphs), either with "-" or maybe numbers for each aim, such as (1), (2), etc.
Reply: As suggested, we have numbered each aim, such as (1) and (2).
3. In the Discussion, it is not necessary to start the paragraphs with "first..., second..., third...". It
will be a more pleasant read
Reply: Thank you for the suggestion. We have accordingly deleted them from the start of the paragraphs
Reviewer 2 Report
This is a well conducted and well written study that is marred by far too many numbers and statistical details of little relevance to the conclusions of the study.
Table 1: The table heading should be changed to: Participant characteristics and their relationship to health prmotion behaviour.
The n column should have number of participants only, delete percentages.
The Means and SD should be deleted. Instead of the 'Mean SD' column, there should be a column showing the health promoting score for each row.
The list of specific disabilities thould be taken out of the Table and better presented in the text, without percentages, and preferably arranged by the number of participants: ...neck/back pain (77), sleep disorders (40),.......and so on until .........other (13).
Line 206: insert 'significant' before 'differences'
Table 2: Similar changes as recommended for Table 1: No percentages, include health promoting scores
Table 3: The correlations between each pair of variables have little interest. Only the last line of the table, the correlatons between health promotion behaviour and each of the other variables, is of interest, only these are referred to in the text. The rest of the table should be omitted.
Line 239: 'Table 3' should be 'Table 4'
Table 4 and the description of models 1 and 2. The details of models 1 and 2 should be omitted. As far as I can see, only the results of Model 3 are basis for the discussion and conclusion, and Table 4 should be limited to the results from model 3.
Author Response
Thank you for the opportunity to revise our manuscript. We appreciate this careful review and your constructive suggestions. The manuscript has improved substantially after making the suggested edits.
Dear Editors:
Thank you for the opportunity to revise our manuscript. We appreciate this careful review and your
constructive suggestions. The manuscript has improved substantially after making the suggested edits.
The detailed responses to each comment follow below.
Response to Reviewer 2 Comments
1. Table 1: The table heading should be changed to: Participant characteristics and their
relationship to health promotion behaviour.
Reply: Thank you for your comment. We have revised the table heading accordingly.
2. Then column should have number of participants only, delete percentages.
Reply: We have deleted the percentages, leaving only the number of participants.
3. The Means and SD should be deleted. Instead of the 'Mean SD' column, there should be a
column showing the health promoting score for each row.
Reply: In accordance with your comment, we have modified the health promotion score for each row. The
Table 1 section has been revised accordingly.
4. The list of specific disabilities should be taken out of the Table and better presented in the text,
without percentages, and preferably arranged by the number of participants: ...neck/back pain
(77), sleep disorders (40),.......and so on until .........other (13).
Reply: We have deleted the specific disabilities in Table 1. We have presented the number of
participants in the text. Deleted parts are indicated in blue. The specific disability table has been
presented as an appendix. The relevant text has been revised as follows:
 In terms of health problems, 118 participants answered that they had health problems, whereas 44 said that they were healthy. We further inquired about the specific disability or health problem that these 118
participants had and found the following: neck/back pain (77), sleep disorders (40), emotional problems
(15), and eye/vision problems (13). In terms of the number of hospital visits per month, 58 participants
answered that they visited the hospital at least once per month.
5. Line 206: insert 'significant' before 'differences'
Reply: We have modified line 206 based on your comment.
6. Table 2: Similar changes as recommended for Table 1: No percentages, include health
promoting scores
Reply: We agree with the reviewer’s suggestion. We have deleted the percentages from Table 2 and
included the health-promoting scores.
7. Table 3: The correlations between each pair of variables have little interest. Only the last line
of the table, the correlatons between health promotion behaviour and each of the other variables,
is of interest, only these are referred to in the text. The rest of the table should be omitted.
Reply: We have modified the variables in Table 3 for clarity. Deleted parts are indicated in blue.
8. Line 239: 'Table 3' should be 'Table 4'
Reply: Thank you for pointing this out. We have made the necessary correction.
9. Table 4 and the description of models 1 and 2. The details of models 1 and 2 should be omitted.
As far as I can see, only the results of Model 3 are basis for the discussion and conclusion, and
Table 4 should be limited to the results from model 3.
Reply: We appreciate the reviewer’s valuable comments. We have deleted the details of Models 1 and 2
from the Results section. The discussion and conclusions are based on Model 3, and the results in Table 4 describe only the results from Model 3.
Round 2
Reviewer 2 Report
There are still changes to be made:
Lines 195 - 199: It must be told that what is shown in the parentheses, are the mean and SD of the health promotion behaviour score.
Lines 199-202: In the first version, eye/vision problems were noted in 14 participants, now in 13. check which is correct. It should be mentioned here that a list of further problems is given in a supplementary Table.
Table 1: The heading of the 4th column should be Health promotion behaviour score
Table 1: The bottom 6 lines of the table are of a different type than the rest of the table and are meaningless with the values now presented. These lines should be presented in a separate table, and with the values presented in the first version of the manuscript
Table 3: The Table is changed as I suggested previously, but the table heading is no longer appropriate. It should now rather be: Correlations between the Health promotion scores and each of the study variables.
Table 4: Although the authors reply is that the table now only shows the results of Model 3, as suggested, in my printout of the revised manuscript, the table is unchanged, still with the outcomes of models 1 and 2
Author Response
Thank you very much for reviewing our study. We have attempted to address the reviewer' comments, and in so doing, we believe we have improved the manuscript substantially. The revisions made to the manuscript are highlighted in blue font.
Dear Editors:
Thank you very much for reviewing our study. We have attempted to address the reviewer' comments, and in so doing, we believe we have improved the manuscript substantially. The revisions made to the manuscript are highlighted in blue font.
Response to Reviewer 2 Comments
- Lines 195 - 199: It must be told that what is shown in the parentheses, are the mean and SD of the health promotion behaviour score.
Reply: Based on this comment, we have added the mean and SD of the health promotion behaviour score mentioned in the 3.1.1. General characteristics section.
- Lines 197-198: The mean and standard deviation of health promotion behavior scores are shown in Table 1.
- Lines 199-202: In the first version, eye/vision problems were noted in 14 participants, now in 13. check which is correct. It should be mentioned here that a list of further problems is given in a supplementary Table.
- LineS 201-205: We further inquired about the specific disability or health problem that these 118 participants had and found the following: neck/back pain (77), sleep disorders (40), emotional problems (15), eye/vision problems (14), and other (13) (See furtherer disabilities or health problems in Supplement Table S1).
- Table 1: The heading of the 4th column should be Health promotion behaviour score
Reply: The heading of the 4th column has been revised as follows:
Variables |
Categories |
n |
Health Promotion Behavior Score (M±SD) |
t |
p |
- Table 1: The bottom 6 lines of the table are of a different type than the rest of the table and are meaningless with the values now presented. These lines should be presented in a separate table, and with the values presented in the first version of the manuscript.
Reply: We appreciate the reviewer’s detailed comments. We have now followed the suggestion from the reviewer and removed the bottom 6 lines of the table 1.
- Table 3: The Table is changed as I suggested previously, but the table heading is no longer appropriate. It should now rather be: Correlations between the Health promotion scores and each of the study variables.
Reply: According to reviewer’s advice, the relevant table heading was corrected as follows.
- Table 3. Correlations Between Health Promotion Score and each of the Study Variables
- Table 4: Although the authors reply is that the table now only shows the results of Model 3, as suggested, in my printout of the revised manuscript, the table is unchanged, still with the outcomes of models 1 and 2
Reply: We thank the reviewer for this important suggestion. As suggested the results of Model 1, 2 have been removed from Tables 4 and a paragraph was revised in the text as follows: (Lines 244-257)
- A multiple regression analysis was conducted to examine the factors affecting nurses’ health promotion behavior during the COVID-19 pandemic. Table 4 shows the results. The Durbin-Watson statistic was used to check for autocorrelation between error terms before the analysis; the results showed that there was no autocorrelation. A normal distribution was assumed for the error terms. On examining whether there is multicollinearity between the independent variables through the tolerance limit and the variance inflation factor, the tolerance limit was .66–.90 (higher than .10), and the variance inflation factor was 1.09–1.57 (less than 10). Thus, there was no multicollinearity between the independent variables.
According to the analysis, the variables affecting health promotion behavior were social support (β =.40, p < .001) and self-efficacy (β = .27, p = .014). In other words, higher social support and self-efficacy were correlated with better health promotion behavior. The adjusted R2 was .514, meaning that the measured variables explained 51.4% of the variance in health promotion behavior.